# Nitrogen dynamics as a function of soil types, compaction, and moisture

**Saurav Das** [1]*, **Ankita Mohapatra** [1,2], **Karubakee Sahu** [1,2], **Dinesh Panday** [3], **Deepak Ghimire** [1], **Bijesh Maharjan** [1]

**1** Department of Agronomy and Horticulture, University of Nebraska, Lincoln, NE, United States of America, **2** Department of Agronomy, Odisha University of Agriculture and Technology, Odisha, India, **3** Rodale Institute–Pocono Organic Center, Long Pond, PA, United States of America

* sdas4@unl.edu

**Data Availability Statement:** Within the manuscript and supporting information files!

**Funding:** The author(s) received no specific funding for this work.

## Abstract

In this study, the complex interactions between soil types, compaction, and moisture on nitrogen (N) transformation processes such as ammonia ($NH_3$) volatilization, ammonification, nitrification, and denitrification were examined over a 30-day period using a simulated column approach. Two soil types: loam, and sandy loam, were subjected to three compaction treatments—control, surface, and sub-surface compaction—and two moisture regimes, dry and wet. Liquid urea ammonium nitrate (32-0-0) was used as the N fertilizer source at a rate of 200 kg N ha$^{-1}$. Key indicators of N transformations were measured, including residual concentrations of ammonium ($NH_4$-N) and nitrate ($NO_3$-N), $NO_3$-N leaching, $NH_3$ volatilization, and nitrous oxide ($N_2O$) emissions. Findings revealed that compaction significantly increased residual $NH_4$-N concentrations in deeper soil profiles, with the highest 190.80 mg kg$^{-1}$ recorded in loam soil under sub-surface compaction and dry conditions. Nitrification rates decreased across both soil types due to compaction, evidenced by elevated residual $NH_4$-N levels. Increased $NO_3$-N leaching was observed in loam soil (178.06 mg L$^{-1}$), greater than sandy loam (81.11 mg L$^{-1}$), due to initial higher residual $NO_3^-$ in loam soil. The interaction of compaction and moisture most affected $N_2O$ emissions, with the highest emissions in control treatments during dry weather at 2.88 kg ha$^{-1}$. Additionally, higher $NH_3$ volatilization was noted in moist sandy loam soil under control conditions at 19.64 kg ha$^{-1}$. These results highlight the necessity of considering soil texture, moisture, and compaction in implementing sustainable N management strategies in agriculture and suggest recommendations such as avoiding broadcast application in moist sandy loam and loam soil to mitigate $NH_3$ volatilization and enhance N use efficiency, as well as advocating for readjustment of fertilizer rate based on organic matter content to reduce potential $NO_3$-N leaching and $N_2O$ emissions, particularly in loam soil.

## 1. Introduction

Agricultural production increased significantly as nitrogen (N) became available for croplands following the end of World War II [1]. However, the widespread use and, oftentimes,

**Competing interests:** The authors have declared that no competing interests exist.

**Abbreviations:** C, Control; SC, Surface Compaction; SSC, Sub-Surface Compaction; SL, Sandy Loam; L, Loam; N, Nitrogen; D, Dry; W, Wet.

mismanagement of N fertilizers have raised questions on the agricultural system's sustainability, given the detrimental effects of N on the environment. Consequently, managing N inputs to achieve simultaneously profitable crop production and minimal environmental implications is crucial for sustainable crop production.

Effective N management requires a comprehensive understanding of intricate biogeochemistry of N within the soil system. Nitrogen is susceptible to losses through various processes, including leaching, volatilization, denitrification, and surface runoff, making it a challenging nutrient for efficient management [2]. Volatilization and denitrification are among the primary contributors to agricultural N losses to the atmosphere. Agricultural practices notably contributed to the steady increases of atmospheric nitrous oxide ($N_2O$) level, with a significant share of 3.9 to 5.3 Tg N $yr^{-1}$. This has led to an increase in $N_2O$ level from 290 ppb in 1940 to 330 ppb in 2017 [3]. Pre-plant fertilization and surface application of dry fertilizers can result in N loss through ammonia ($NH_3$) volatilization and consequent air quality issues [3–5]. The transformation of N in the soil plays a critical role in determining N availability and its fate, nitrate ($NO_3$) loss to water bodies, and $N_2O$ and $NH_3$ emissions. Nitrate leaching is a concern in many parts of world for groundwater contamination and the negative human health impact associated with it. High levels of $NO_3$ in surface water bodies can cause eutrophication and biodiversity loss. Therefore, it is essential to develop effective N management strategies that account for the complex N dynamics in the soil system to mitigate environmental implications. Efficient and eco-friendly N management is challenging due to multifaceted interactions, including humans, plants, microbes, and livestock. The highly reactive nature of N increases the probability of its loss from soil to rivers, lakes, and the atmosphere. Therefore, understanding how agronomic management practices and associated issues affect N dynamics is essential. Gaining insights into the biogeochemistry of N under different management and soil conditions is important for developing sustainable, environmentally friendly agronomic practices and for safeguarding both ecosystems and food production systems.

Nitrogen transformation is a complex interplay between agronomic practices and environmental factors. Among many effects of farming practices, soil compaction has emerged as a pervasive issue in modern agriculture due to the extensive use of heavy machinery on farmland. Factors contributing to soil compaction encompass excessive machinery usage, intensive cropping systems, inadequate crop rotation, intensive grazing, and suboptimal land management practices [6]. Compaction reduces pore spaces, particularly large pores, resulting in reduced porosity and soil air content. The degree of soil compaction is influenced by soil moisture content. The use of soil (ground) engaging tools, equipment, and heavy machinery during high soil moisture content can significantly increase soil compaction [7]. Given the direct relationship between pore space and water movement, compacted soil can obstruct water infiltration and drainage, and subsequently affect the N transformation and movement of N in the soil profile [8]. Compaction may either increase or reduce N loss, contingent upon the affected N transformation pathways [9]. Soil compaction presents a significant challenge to today's agriculture, causing degradation, reduced soil productivity, and increased soil erosion and runoff [10]. Therefore, comprehending the N cycle in compacted soil environments across diverse climatic conditions and soil types is crucial for efficient N management.

Several previous studies such as those conducted by Blumfield et al., (2005) [11], Jensen et al., (1996) [12], and Longepierre et al., (2022) [13], have contributed valuable insights into the influence of soil compaction on N-transformation and loss within the soil environment. Blumfield et al., (2005) [11] concluded that soil compaction did not significantly affect N mineralization. Conversely, both Jensen et al., (1996) [12] and Longepierre et al., (2022) [13] reported a reduction in the rate of N-mineralization due to compaction. However, these contributions, while noteworthy, do not provide a comprehensive understanding of the multi-

faceted N cycle under varying degree of soil compaction, soil types, and environmental conditions. Soil types, particularly their variable textural and physical properties, can influence water storage and infiltration and variably affect the microbial community structure and activity. For instance, sandy soil, with larger particles and more pore space, have higher infiltration rate, and which significantly affect the mobilization of reactive N. Soils with finer texture, such as clay, which have higher organic matter storage potential, provide substrates to microbes, enhancing nitrogen transformation. Furthermore, during dry weather, microbial activity is significantly reduced due to lack of moisture, impacting nitrogen transformation pathways [14–16]. Thus, it is critical to understand the interactive nature of compaction, soil type, and moisture on N dynamics.

To address this knowledge gap, the present study uses a controlled environmental laboratory simulation using soil columns to evaluate how surface and subsurface compaction affect nitrogen transformation processes such as ammonia volatilization, nitrification, and denitrification in dry and wet weather conditions and in loam and sandy loam soils. This understanding will be instrumental in developing more effective soil management strategies, which can optimize N use efficiency and minimize environmental impacts, thereby contributing to sustainable agricultural practices.

## 2. Materials and methods

### 2.1 Description of soil

Soils were collected from 0 to 20 cm depth, representing the plow layer, from the University of Nebraska–Lincoln (UNL) Panhandle Research, Extension, and Education Center (PREEC) in Scottsbluff, Nebraska (41˚53'31.0"N, 103˚40'54.8"W) and the UNL High Plains Agricultural Lab (HPAL) in Sidney, Nebraska (41˚14'10.6"N, 102˚59'36.9"W) in spring 2020. Soil from PREEC was sandy loam (Tripp soil series: sand ~60%, silt ~20%, and clay ~20%), and soil from HPAL was loam (Duroc soil series: clay content ~ 30%, silt ~ 50%, and sand ~ 20%). The average precipitation in this region ranges from 355 mm to 430 mm. The baseline soil properties are provided in **S1 Table**. The collected soil was cleaned by removing stubbles and other residues, air-dried (at room temperature), and then sieved through a 2 mm mesh. When soils were packed in columns, gravimetric water content (GWC) was maintained at 10% by adding water, corresponding to 70% and 50% of field capacities for sandy loam and loam, respectively. Sandy loam soils have larger particle sizes and, consequently, larger pore spaces. They drain more quickly and have lower field capacities than finer-textured soils. By choosing 70% field capacity for sandy loam, the study aims to simulate a moisture level that is sufficient for plant growth and soil microbial activity without over-saturating the soil. Loam soils have a more balanced mix of sand, silt, and clay, leading to a moderate water-holding capacity and better nutrient retention than sandy soils. Choosing 50% of field capacity for loam reflects an optimal balance between ensuring enough moisture for plant and microbial activities and avoiding waterlogging, which can lead to reduced soil aeration. It is important to note that the loam soil collected for this experiment had a relatively higher amount of residual nitrate than the sandy loam soil (**S1 Table**).

### 2.2 Soil column setup and treatment

Soils were packed to a height of 24 cm in clear acrylic columns of 5 cm diameter (Fig 1). To prevent plates from getting clogged with soil, porous ceramic plates (0.1 MPa) were topped with Whatman no. 42 filter paper at the bottom of each soil column (Fig 1). The experiment had three soil compaction treatments including i) the control (C), which had a uniform bulk density of 1.3 and 1.4 kg m$^{-3}$ in the soil column for loam and sandy loam, respectively; ii)

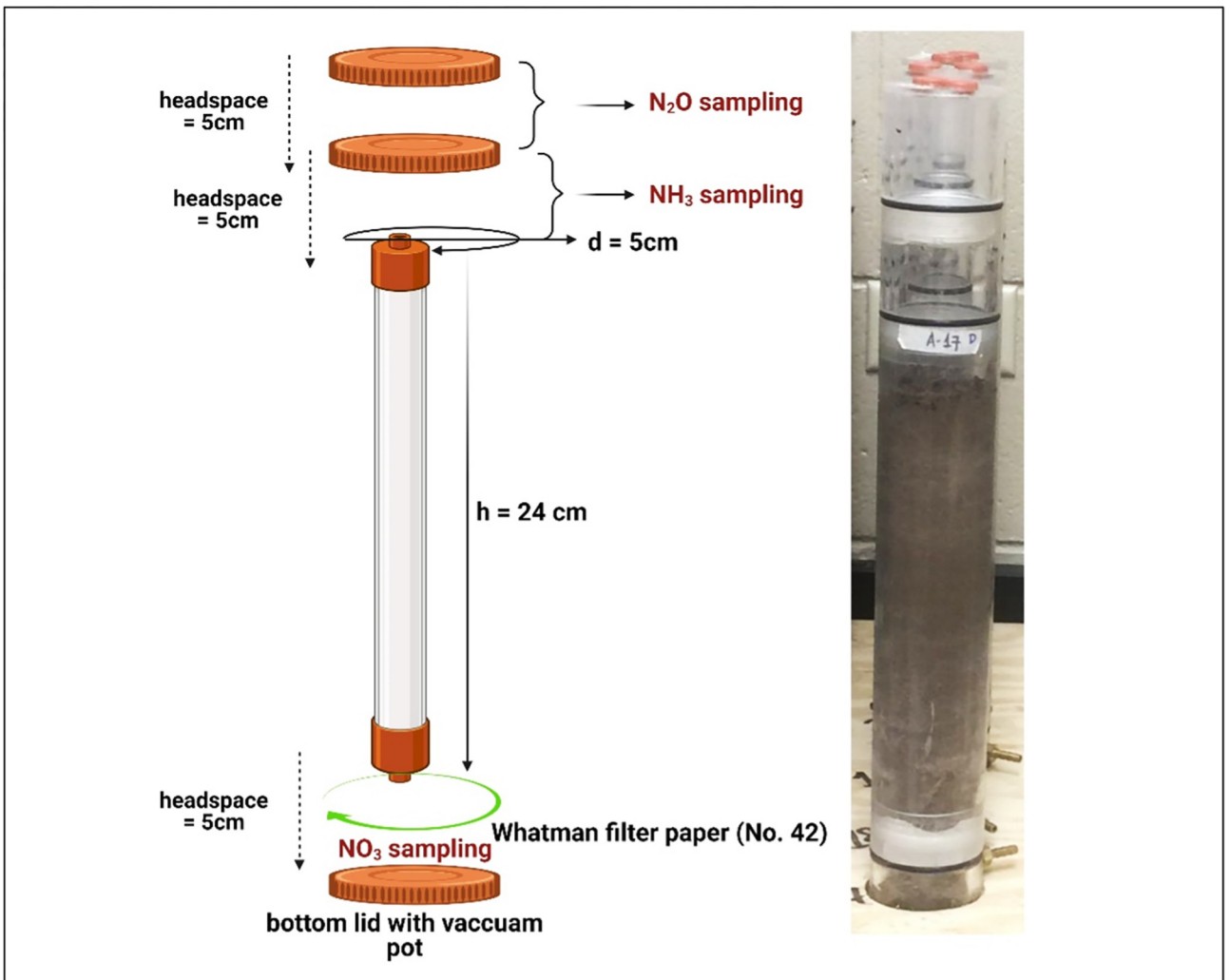

**Fig 1. Soil column structure and description of soil sampling ports.** Here d = diameter and h = height.

surface compaction (SC), which was achieved during soil packing in columns with a targeted bulk density of 1.5 and 1.6 kg m$^{-3}$ on the top 10 cm depth for loam and sandy loam, respectively; and iii) sub-surface compaction (SSC), which had a bulk density of 1.5 and 1.6 kg m$^{-3}$ at the bottom 10 cm depth for loam and sandy loam, respectively. The soil layers in columns other than the compacted layers in both SC and SSC treatments had bulk densities of 1.3 and 1.4 kg m$^{-3}$ for loam and sandy loam, respectively. Each column had lids at either end. The bottom lid had a vacuum port to aid in the collection of leachates (Fig 1). The pump (0.25-horse-power air motor) was connected to the vacuum port to apply suction, which facilitated the passage of water through the ceramic plate. The top lid had two parts: the lower one was 5 cm tall and threaded onto the main column, and the upper lid installed acid traps for NH$_3$. The upper lid was also 5 cm tall, with a closed-end fitting with two septum ports used for N$_2$O gas sampling (Fig 1).

Urea Ammonium Nitrate (32-0-0), a liquid N fertilizer (density = 1.36 g ml$^{-1}$), was added to each soil column at 90 µl using a pipette in a dropwise manner, and thoroughly mixed in the upper 2 cm. Thus, each soil column received 39.25 x 10$^{-3}$ g N fertilizer, equivalent to 200 kg N

ha$^{-1}$. We used 200 kg N ha$^{-1}$ to match the 100% N recommendation for the yield goal of 150 to 170 bushels acre$^{-1}$ for dryland corn. The experiment was a three-factorial design that included two soil types (sandy loam and loam), three soil compaction levels (control, SC, and SSC), and two moisture regimes (dry and wet), resulting in 12 combinations in four replications, totaling 48 columns. The experiment was conducted for 30 days. Water was added daily to the soil columns in the dry regime to simulate the average rainfall (1981 to 2010) in Scottsbluff, NE, in May, totaling 65.02 mm. The water was added using a pipette. The May precipitation in 2019, the recent wettest year in Scottsbluff, NE (128.78 mm), was considered for the wet regime in the experiment (S2 Table). All soil columns were maintained at a constant room temperature.

## Calculations

*A. Required Rate of fertilizer per column (at 200 kg N ha$^{-1}$)*:

 *I. Diameter of the column = 5cm, so the radius is = 2.5 cm*

 *II. Surface area of the soil column, $A = \pi r^2 = 3.14 \times 2.5 \times 2.5 = 19.625 \, cm^2$*

 III. Rate of fertilizer (per cm$^2$) = 200 kg N /ha = (200 x 1000 g) / 1 x 10$^8$ cm$^2$ = 2 x 10$^{-3}$ g N per cm$^2$

 IV. Thus, each column should receive N = area of the column x rate of fertilizer per cm$^2$

$$= 19.625 \times (2 \times 10^{-3} \, g \, N \, per \, cm^2)$$
$$= 39.25 \times 10^{-3} \, g \, N \, each \, column$$

 *B. UAN needed for this required rate*:

*Nitrogen in UAN = 32% and density = 1.36 g ml$^{-1}$*

*Total weight of UAN needed = $(39.25 \times 10^{-3})/0.32 = 122.656 \times 10^{-3}$ g*

*Volume of UAN = weight of UAN / density of UAN*
$$= (122.656 \times 10^{-3} \, g) / 1.36 \, g \, ml^{-1}$$
$$= 0.090 \, ml = 90 \, \mu l$$

## 2.3 Sample collection and chemical analysis

The NH$_3$ volatilization was measured using the acid trap method [17]. A sponge of 5 cm diameter and 1.3 cm thickness was soaked with 5 ml of H$_3$PO$_4$-Glycerol solution (40 ml glycerol, 50 ml H$_3$PO$_4$ acid, and 910 ml deionized water) as a trapping media for NH$_3$. Traps were placed in the lower part of the top lid of each column. As the N losses through volatilization primarily happens in the initial seven to fourteen days following the fertilizer application and gradually decreases after that, we have arranged our sampling dates accordingly. In the first two weeks, we collected every alternate day and followed every fourth day after that. The traps were placed on day 0 and replaced with new ones on days 1, 2, 3, 5, 7, 9, 11, 13, 17, 21, 25, and 29. The NH$_3$ in acid traps was extracted with 2M KCl, brought the total extractant volume to 50 mL by adding 2M KCl, and analyzed for NH$_4$-N using the flow injection method [18]. The concentration of NH$_3$ i.e., mg L$^{-1}$ was converted to kg ha$^{-1}$ by multiplying with extraction volume and extrapolating the area.

The N$_2$O gas samples (25 mL) were collected using a 35-mL syringe from the septum port on the upper lid at 0 and 30 minutes and transferred into 25-mL air-evacuated glass vials. During the gas sampling, the NH$_3$ traps were removed. The 0-minute samples were taken after removing the acid traps and before closing the lid. The samples were collected on days 1, 3, 5, 7, 9, 11, 13, 15, 17, 19, 21, 23, 25, and 29. Gas samples were analyzed for N$_2$O using gas chromatography (450-GC, Varian) and an electron capture detector [19,20].

For leachate sample collection, suction was applied to soil columns using a vacuum pump (3 cubic feet per minute two-stage vacuum pump, Pittsburgh Automotive, PA, USA). The outlet of the pump was fitted to the suction port in the bottom lid and allowed to run for 90

seconds to facilitate the drainage of water collected through the porous ceramic plate into the bottom lid of the column. Periodically (every alternate day), attempts were made to collect leachates, and leachates were successfully collected on days 24, 25, 27, 28, and 30. The flow injection method was used to analyze the leachate samples for $NO_3$-N [18].

For residual $NO_3$-N and $NH_4$-N, on the 30[th] day, the soil column was broken into five segments, four with an increment of 5 cm and the fifth one with 4 cm. Residual $NH_4$-N and $NO_3$-N was extracted using 2M KCl solution and analyzed using a flow analyzer [18]. Note that the columns' caps were periodically opened and closed during the sampling event, which ensured there was no oxygen limitation.

Soil pH was assessed by mixing soil samples with water at a 1:2 soil-to-water ratio. The mixture was then stirred to ensure uniform suspension, and pH levels were measured using a calibrated pH meter, providing an accurate representation of soil acidity. Soil organic matter content was determined using the loss-on-ignition (LOI) method. For total carbon analysis, the dry combustion method was employed. Micronutrient analysis for Zinc (Zn), Iron (Fe), Manganese (Mn), and Copper (Cu) was conducted using the DTPA (diethylenetriaminepentaacetic acid) extraction method and then quantified using atomic absorption spectrophotometry [21]. Cation analysis, including Calcium (Ca), Magnesium (Mg), Potassium (K), and Sodium (Na), was performed using the ammonium acetate extraction method, which were then measured using inductively coupled plasma optical emission spectrometry (ICP-OES) [22].

## 2.4 Data analysis

The $N_2O$ concentration values were converted to mass per volume using the universal gas law equation. Daily gas flux rates ($mg\ m^{-2}\ min^{-1}$) were calculated as the linear or quadratic change in headspace $N_2O$ concentration based on regression analysis with the highest $R^2$ value. Trapezoidal integration was used to calculate the total $N_2O$ emission for each treatment column. To evaluate the significance of treatments on $NO_3$ leachate, $N_2O$ emission, $NH_3$ volatilization, and residual $NO_3$ and $NH_4$, an analysis of variance (ANOVA) was conducted in R using the "aov" function. For $N_2O$ and $NH_3$ emissions, a cumulative mean ($kg\ ha^{-1}$) was calculated, and the ANOVA was performed with the cumulative mean value. Fischer Least Square Difference (LSD) was performed to evaluate the mean separations for each individual treatment and interaction using the "agricolae" package and "LSD.test" function. A p-value adjustment test was done using Bonferroni correction to protect from the type-1 error.

The data are represented as the mean of four independent replications ± standard deviation. The values from soil column fragments for each individual column were averaged to calculate the mean and standard deviation. For residual $NO_3$-N and $NH_4$-N, the depth was also used as a predictor variable to determine the distribution across the soil profile and their differences. The normality assumption of residuals was tested using qqplot.

## Calculations steps for N2O emissions

I. The number of moles of the gas was calculated using the universal gas law equation:

$$PV = nRT \text{ i.e., } n = (P\ x\ V)/(R\ x\ T)$$    Eq1

where, n = Number of moles of the examined gas
 P = Atmospheric pressure (Pa)
 R = Universal Gas constant ($m^3\ Pa\ mol^{-1}\ K^{-1}$)
 T = Temperature (K) [273.15 + t˚C], t is the measured room temperature
 V = Volume of $N_2O$ gas in the chamber ($m^3\ min^{-1}$)
 = $\Delta C$ ($ppm\ min^{-1}$) x $10^{-6}$ x Total headspace volume, $V_{headspace}$ ($m^3$)

*ΔC is the rate of change $N_2O$ in concentration per unit of time (ppm $hr^{-1}$) and was calculated as the linear slope of $N_2O$ concentration (ppm $min^{-1}$) between two measurement points (0 min and 30 min)*

1. II. The number of moles of the gas, n (mol $N_2O$ $min^{-1}$) obtained from Eq 1 was then multiplied by the molecular weight of $N_2O$-N to obtain the flux (g $N_2O$-N $min^{-1}$) from the column headspace.

*i.e. Flux from the column headspace (g $N_2O$-N $min^{-1}$) = n (mol $N_2O$ $min^{-1}$) x molecular weight of $N_2O$-N (g $mol^{-1}$)*

2. III. The Flux (g $N_2O$-N $min^{-1}$) was then converted to per unit area basis (g $N_2O$-N $m^{-2}$ $min^{-1}$) by dividing it by the surface area of the headspace (A).

*i.e. Flux (g $N_2O$-N $m^{-2}$ $min^{-1}$) = Flux (mol $N_2O$ $min^{-1}$) / Surface area of the headspace, A ($m^2$)*
*And Flux (g $N_2O$-N $m^{-2}$ $hr^{-1}$) = Flux (g $N_2O$-N $m^{-2}$ $min^{-1}$) x 60*

3. IV. Cumulative $N_2O$ Emission (kg $ha^{-1}$) = Σ [0.5 * (Previous day flux + Current day flux) *(Current date–Previous date) *Unit Conversion factor (kg $ha^{-1}$ $day^{-1}$)]

*Given the unit of flux is in kg $ha^{-1}$ $day^{-1}$, the unit conversion factor is 0.24 [1 g = $10^{-3}$ kg; 1 ha = $10^4$ $m^2$; 1 day = 24 hour] and 0.5 is the coefficient calculated from Trapezoid integration (area under curve value)*

## 3. Results

### 3.1 Nitrate ($NO_3$-N) leaching

Soil types (sandy loam and loam) and moistures (dry and wet) significantly affected $NO_3$-N leaching (Table 1). A statistically significant difference in $NO_3$-N leaching was observed between the two soil types, with loam soil showing higher levels of leaching than sandy loam. Similarly, a higher $NO_3$-N leaching was observed in wet weather compared to dry weather

**Table 1. Summary table for analysis of variance[&].**

| Treatments Factors (p-value) | Leachate | Gas Emissions | | Soil Residual Mineral N | |
|---|---|---|---|---|---|
| | $NO_3$-N (mg $L^{-1}$) | $N_2O$ (kg $ha^{-1}$) | $NH_3$ (kg $ha^{-1}$) | $NO_3$-N (mg $kg^{-1}$) | $NH_4$-N (mg $kg^{-1}$) |
| **Soil Type** | **<0.001**[§] | 0.765 | **<0.001** | **<0.001** | 0.50 |
| Loam | 178.06 ± 173.94a[ø] | 1.32 ± 1.35 | 14.98 ± 1.82 b | 28.05 ± 22.98 | 54.19 ± 72.67 |
| Sandy loam | 81.11 ± 82.10 b | 1.26 ± 0.66 | 15.93 ± 3.27 a | 4.91 ± 3.41 | 59.41 ± 39.58 |
| **Moisture** | **<0.001** | **<0.001** | **<0.001** | **<0.001** | **< 0.001** |
| Dry | 21.20 ± 13.20 b | 1.61 ± 1.38 a | 13.85 ± 2.21 b | 21.39 ± 24.49 a | 85.95 ± 66.66 a |
| Wet | 148.98 ± 137.14 a | 0.97 ± 0.39 b | 17.07± 2.05 a | 11.57 ± 12.99 b | 27.65 ± 25.40 b |
| **Compaction** | 0.131 | **0.003** | **<0.001** | **< 0.001** | **< 0.001** |
| Control | 104.54 ± 130.84 | 2.05 ± 1.35 a | 17.39 ± 1.58 a | 24.95 ± 24.26 a | 38.03 ± 45.92 b |
| Surface | 146.69 ± 147.77 | 0.99 ± 0.55 b | 14.59 ± 3.04 b | 18.54 ± 21.40 a | 46.02 ± 47.68 b |
| Sub-surface | 102.35 ± 115.42 | 0.82 ± 0.63 b | 14.40 ± 2.13 c | 5.96 ± 3.21 b | 86.36 ± 68.62 a |
| **Compaction*Soil Type[¶]** | 0.146 | **0.007** | **<0.001** | **< 0.001** | **< 0.001** |
| **Compaction*Moisture** | 0.906 | **0.004** | **<0.001** | 0.072 | **0.007** |
| **Soil Type*Moisture** | 0.143 | 0.742 | **<0.001** | 0.054 | 0.166 |
| **Compaction*Soil Type*Moisture** | 0.819 | 0.059 | **<0.001** | 0.059 | **0.001** |

[&]Values are presented as the means ± standard deviations.

[§]Significant p-value (<0.05) are represented as bold letters.

[¶]All the two-way interactions are presented in Table 2 and three-way interactions are in Table 3.

[ø] Small letter alphabets define the significance in the least square difference (LSD) test, and difference in alphabets defines the significant difference.

(Table 1). Compaction and the interaction of compaction with moisture and soil types did not have any significant effect on $NO_3$-N leaching (Table 1).

## 3.2 Nitrous oxide ($N_2O$) emissions and ammonia ($NH_3$) volatilization

The $N_2O$ emissions were significantly influenced by compaction (p<0.001), moisture (p = 0.003), soil * compaction (p = 0.007), and moisture * compaction (p = 0.004) (Table 1). Higher $N_2O$ emissions were observed in dry weather compared to wet weather (Table 1). Compaction had a significant impact on $N_2O$ emissions, highest emission was recorded in the control treatment, followed by surface compaction and sub-surface compaction. The interaction of compaction * soil types and compaction * moisture also significantly affected $N_2O$ emissions. The maximum emission was observed in loam soil under the control treatment (C x L) and the lowest was observed in sub-surface compaction in loam soil (SSC x L) (Table 2). Similarly, the highest $N_2O$ emission was recorded in control treatment in dry weather (C x D), and the effect of other treatments were at par with that in SSC x D (Table 2).

Ammonia volatilization was significantly affected by soil type, compaction, moisture, and a three-way interaction of compaction * soil * moisture (Table 1). Across all the interactions, ammonia volatilization ranged from 10.84 ± 0.03 kg ha$^{-1}$ to 19.64 ± 0.06 kg ha$^{-1}$. The highest ammonia volatilization was observed in the control treatment under sandy loam soil in wet

**Table 2. Two-way interaction data.** Significant two-way interactions from Table 1, are represented here[&].

| Two-way interactions | $N_2O$ emission (kg ha$^{-1}$) | $NH_3$ volatilization (kg ha$^{-1}$) | Residual $NO_3$-N (mg kg$^{-1}$) | Residual $NH_4$-N (mg kg$^{-1}$) |
|---|---|---|---|---|
| **Compaction*Soil Type[‡]** | | | | |
| C x L | 2.56 ± 1.65 a[ø] | 16.52 ± 1.21 b | 43.96 ± 20.73 a | 2.65 ± 0.79 e |
| C x SL | 1.54 ± 0.78 ab | 18.27 ± 1.46 a | 5.93 ± 2.34 c | 73.40 ± 40.72 b |
| SC x L | 0.87 ± 0.72 b | 12.97 ± 1.37 f | 32.99 ± 21.97 b | 32.49 ± 51.95 d |
| SC x SL | 1.11 ± 0.33 b | 16.22 ± 3.44 c | 4.08 ± 4.60 c | 59.55 ± 41.88 bc |
| SSC x L | 0.53 ± 0.26 c | 15.46 ± 0.17 d | 7.20 ± 3.01 c | 127.43 ± 70.36 a |
| SSC x SL | 1.12 ± 0.77 b | 13.32 ± 2.65 e | 4.71 ± 3.10 c | 45.29 ± 35.83 cd |
| **p-value** | **P = 0.007[§]** | **P <0.001** | **P <0.001** | **P < 0.001** |
| **Compaction*Moisture** | | | | |
| C x D | 2.88 ± 1.49 a | 16.14 ± 0.81 c | 34.17 ± 29.50 | 55.85 ± 57.85 bc |
| C x W | 1.22 ± 0.36 b | 18.64 ± 1.06 a | 15.72 ± 14.01 | 20.20 ± 21.11 d |
| SC x D | 1.13 ± 0.75 b | 12.34 ± 0.70 f | 23.52 ± 25.47 | 68.56 ± 56.96 b |
| SC x W | 0.85 ± 0.23 b | 16.84 ± 2.77 b | 13.56 ± 16.60 | 23.48 ± 21.58 d |
| SSC x D | 0.82 ± 0.81 b | 13.07 ± 2.39 e | 6.48 ± 2.42 | 133.44 ± 63.66 a |
| SSC x W | 0.83 ± 0.45 b | 15.71 ± 0.11 d | 5.43 ± 396 | 39.28 ± 31.15 cd |
| **p-value** | **P = 0.004** | **P <0.001** | P = 0.072 | **P = 0.007** |
| **Soil * Moisture** | | | | |
| L x D | 1.67 ± 1.84 | 14.12 ± 1.80 c | 35.94 ± 27.90 | 77.96 ± 93.21 |
| L x W | 0.96 ± 0.38 | 15.83 ± 1.46 b | 20.16 ± 13.78 | 30.42 ± 33.50 |
| SL x D | 1.54 ± 0.76 | 13.58 ± 2.61 d | 6.84 ± 3.76 | 93.94 ± 21.57 |
| SL x W | 0.97 ± 0.41 | 18.29 ± 1.83 a | 2.98 ± 1.44 | 24.89 ± 12.50 |
| **p-value** | P = 0.742 | **P <0.001** | P = 0.054 | P = 0.166 |

[‡]Here, C = Control, SC = Surface compaction, SSC = Sub-surface compaction, L = Loam soil, SL = Sandy Loam, D = Dry weather, and W = Wet weather.

[&]Values are represented as the means ± standard deviation.

[§]Significant p-value (<0.05) are represented as bold letters.

[ø] Small letter alphabets define the significance in the least square difference (LSD) test, and difference in alphabets defines the significant difference.

**Table 3. Three-way interaction data.** Significant three-way interactions from Table 1, are represented here.

| Three-way interactions[‡] | Residual ammonium (mg kg$^{-1}$) | Ammonia volatilization (kg ha$^{-1}$) |
|---|---|---|
| C x L x D | 2.26 ± 0.51 g[ø] | 15.38 ± 0.08 g |
| C x L x W | 3.05 ± 0.88 g | 17.65 ± 0.04 c |
| C x SL x D | 109.44 ± 1.13 b | 16.90 ± 0.03 d |
| C x SL x W | 37.35 ± 15.98 efg | 19.64 ± 0.06 a |
| SC x L x D | 40.84 ± 70.90 def | 11.69 ± 0.04 j |
| SC x L x W | 24.13 ± 32.93 fg | 14.25 ± 0.05 h |
| SC x SL x D | 96.28 ± 22.23 bc | 12.99 ± 0.02 i |
| SC x SL x W | 22.82 ± 1.10 fg | 19.43 ± 0.08 b |
| SSC x L x D | 190.80 ± 19.42 a | 15.31 ± 0.09 g |
| SSC x L x W | 64.07 ± 21.66 cde | 15.61 ± 0.03 f |
| SSC x SL x D | 76.09 ± 17.58 bcd | 10.84 ± 0.03 k |
| SSC x SL x W | 14.50 ± 12.55 fg | 15.81 ± 0.06 e |

[‡]Here, C = control, SC = surface compaction, SSC = sub-surface compaction, L = loam, SL = sandy loam, W = wet weather, D = dry weather.

[ø] Small letter alphabets define the significance in the least square difference (LSD) test, and difference in alphabets defines the significant difference.

weather (C x SL x W), followed by surface compaction, sandy loam, and wet weather (SC x SL x W) (Table 3). The lowest ammonia volatilization was observed in sub-surface compaction, sandy loam, and dry weather (SSC x SL x D) (Table 3). Significant effects were observed for the day of measurement in NH$_3$ volatilization and N$_2$O emissions. The NH$_3$ volatilization initially increased until the 2$^{nd}$ day for dry weather and the 5$^{th}$ day for wet weather, followed by a gradual decline throughout the experimental period, maintaining a similar pattern both in dry and wet weather (Fig 2A). The N$_2$O emissions in both dry and wet weather followed a comparable pattern, with emissions increasing until the 9th day, decreasing until the 12$^{th}$ day, then a sudden spike on the 15$^{th}$ day, and subsequently declining until the end of the experiment (Fig 2B).

## 3.3 Soil residual ammonium (NH$_4$-N) and nitrate (NO$_3$-N)

Residual NH$_4$-N concentration was responsive to three-way interaction between compaction * soil type * moisture. Higher residual ammonium was recorded in SSC in loam soil and dry weather (SSC x L x D) followed by control x sandy loam x dry weather (C x SL x D) (Table 3). Two-way interaction effects of compaction * soil type, and compaction * moisture, also significantly affected the residual NH$_4$-N concentration. On comparison of compaction * soil type, the highest residual NH$_4$-N was observed for SSC in loam soil, while the lowest was found in loam soil under control treatment (Table 2). Comparing the effect of compaction * moisture, the highest residual NH$_4$-N was recorded in SSC under dry weather conditions (SSC x D), and the lowest was observed in wet weather under control treatment (C x W) and surface compaction (SC x W) (Table 2).

A four-way interaction effect on residual NH$_4$-N was observed involving depth, compaction, soil types, and moisture. In the loam soil in the control treatment, under both dry and wet weather conditions, no difference in residual NH$_4$-N was detected across the soil profile, with concentrations remaining below 2 mg kg$^{-1}$. However, a significant residual concentration of NH$_4$-N was found across the soil profile in sandy loam soil under control treatment in both dry and wet weather, where dry weather had considerably higher concentrations. In SC during

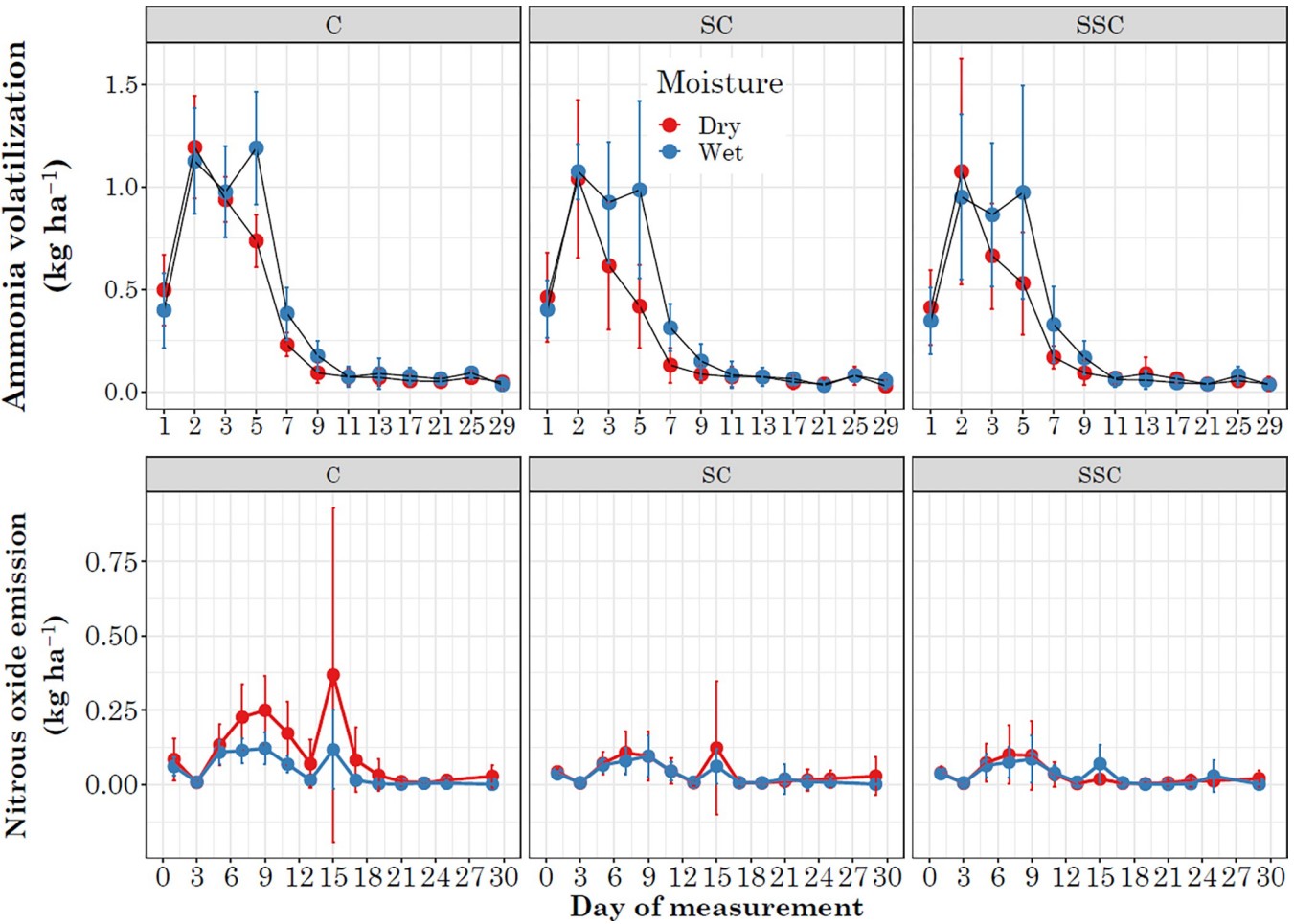

**Fig 2.** Day-wise distribution of ammonia volatilization (A) and nitrous oxide emission (B). Red color represents dry weather and blue color represents wet weather. Here, C = control, SC = surface compaction, SSC = sub-surface compaction.

dry weather, significant residual $NH_4$-N was observed across the sandy loam soil profile. In SSC, the highest residual $NH_4$-N was recorded in loam soil in both dry and wet weather, with the maximum residual ammonium observed at a depth of 24 cm in SSC under dry weather conditions in loam soil (Fig 3).

The results showed a significant effect of soil type, moisture, compaction, and the interaction between compaction and soil type on residual $NO_3$-N concentration (Table 1). Loam soil in the control treatment (C x L) exhibited the highest residual $NO_3$-N concentration, followed by surface compaction and loam (SC x L) (Table 2). On analyzing the effect of depth as a parameter with treatment groups, a significant interaction effect of compaction, soil type, and depth was observed. In sandy loam, no significant differences in residual $NO_3$-N were observed in relation to compaction treatment and depth. Contrary, in loam soil under the control treatment, the highest residual $NO_3$-N was detected at 24 cm. Similarly, in SC, a gradual increase in residual $NO_3$-N was observed with a gradual increase in depth. In SSC, no significant differences in residual $NO_3$-N were observed for both loam and sandy loam across soil depths (Fig 4).

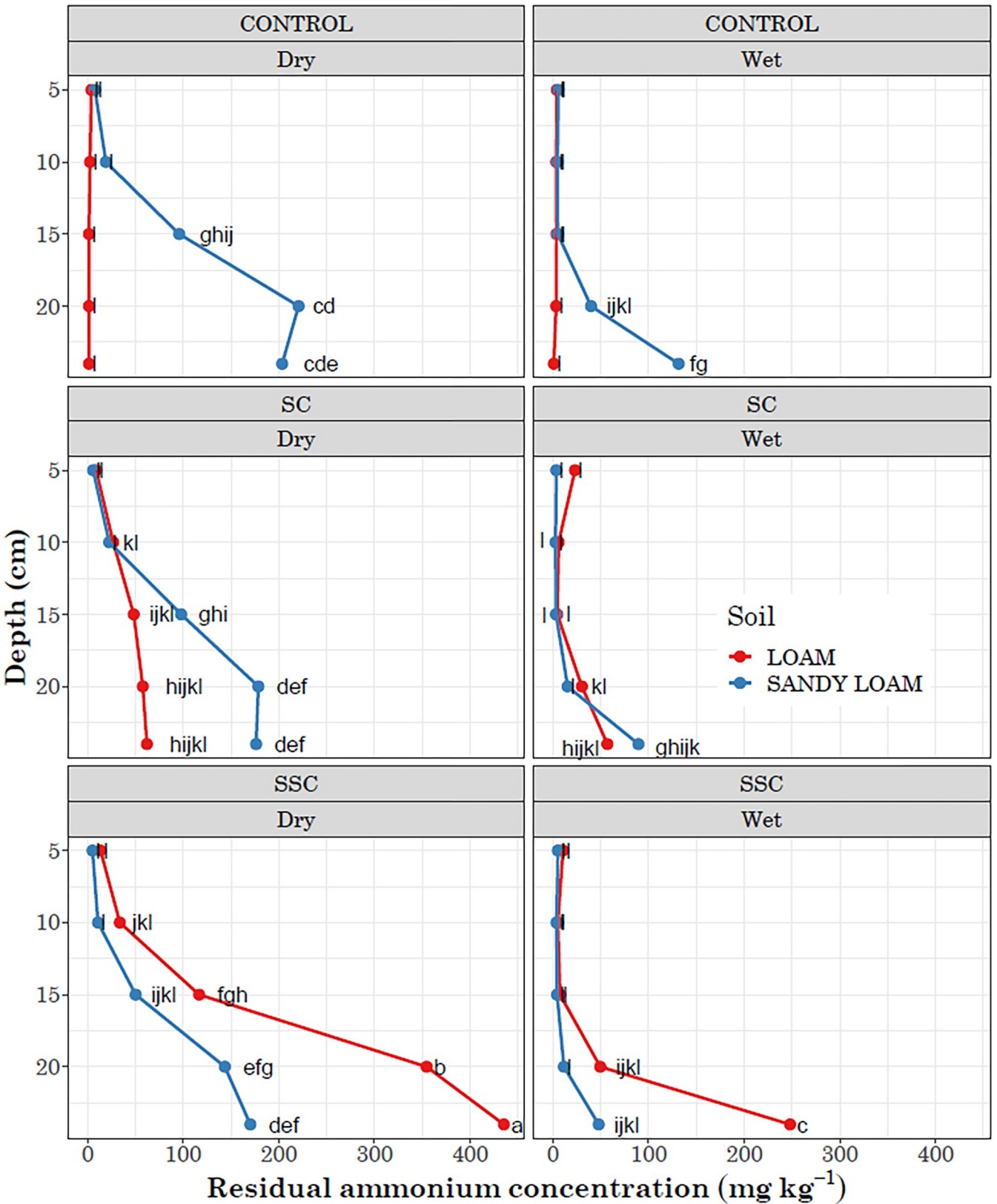

**Fig 3. Distribution of residual ammonium concentration by depth and comparisons between Surface Compaction (SC) and Sub-Surface Compaction (SSC) under simulated dry and wet weather conditions.** The lowercase letters indicate results from a least square difference (LSD) test for multiple mean comparisons. Distinct letters represent significant differences among means, whereas identical letters indicate no significant differences at a p<0.05 significance level.

## 4. Discussion

Sustainable management of N is imperative for both agricultural productivity and environmental quality preservation. As the most critical nutrient for crop yield, N simultaneously functions as a principal contributor to environmental contamination due to its highly reactive nature. Soil compaction has surfaced as a paramount challenge within contemporary intensified agricultural systems, where employing heavy machinery is essential. The interaction between compaction, varying moisture levels, and distinct soil compositions constitutes a significant anthropogenic influence on N transformation processes.

The $NH_3$ volatilization is a process in which ammonia is released into the atmosphere from the soil, particularly when soils are moist and warm, and the source of urea is on or near the soil surface [23–25]. The process occurs when $NH_4$ is converted to $NH_3$ gas at the soil surface and is then transported to the atmosphere. Urea application to a wet soil surface results in

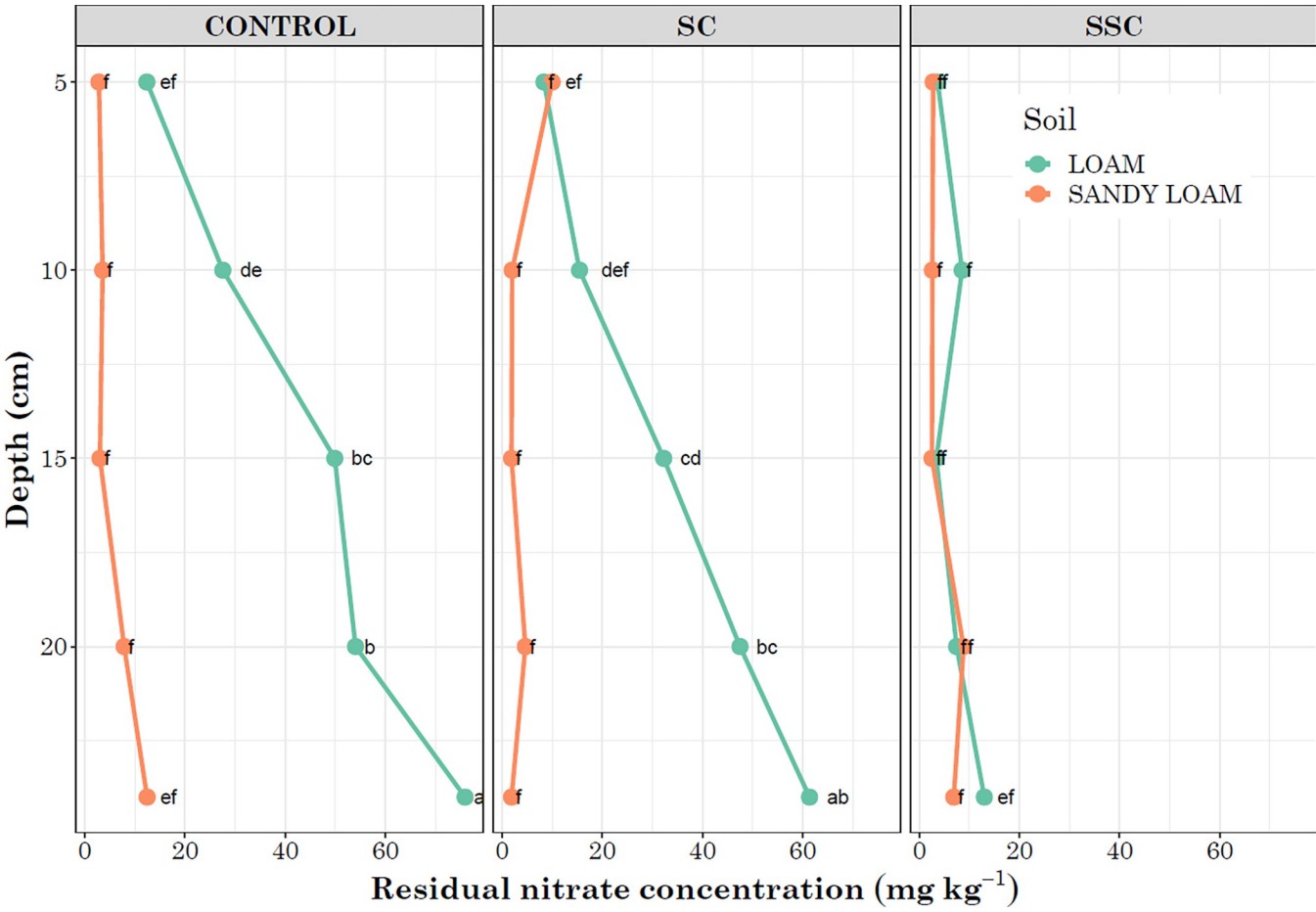

**Fig 4. Depth-wise distribution of residual nitrate concentration.** Here SC = surface compaction and SSC = sub-surface compaction. The lowercase letters indicate results from a least square difference (LSD) test for multiple mean comparisons. Distinct letters represent significant differences among means, whereas identical letters indicate no significant differences at a p<0.05 significance level.

increased volatilization as the rate of hydrolysis increases and movement into the soil decreases due to the amount of water-filled pore spaces [23]. Urea ammonium nitrate (UAN) used in this study is a liquid nitrogen fertilizer containing a mixture of urea and ammonium nitrate. It is commonly used in agriculture due to its ease of handling, storage, and application. However, $NH_3$ volatilization is a concern when applying UAN, as it can lead to N loss and reduced N use efficiency. Broadcast application (or surface spray) in soil with a significant amount of initial moisture can increase $NH_3$ volatilization as found in our simulated study (Table 1). Interaction of compaction treatments with moist soil (wet weather) potentially increased $NH_3$ volatilization for both loam and sandy loam soils (Table 3). Several studies have also recorded application of urea-based fertilizer in moist soil with slow drying can increase $NH_3$ volatilization [23,26]. A study conducted in Montana observed that 30 to 40% of the applied N in the form of urea was lost as volatilization when applied to a moist soil surface [27]. Similarly, in an Oregon-based trial, over 60% of the applied N was lost when urea was administered to freshly irrigated soil without any subsequent irrigation, and only light, scattered rainfall occurred during the following 24-day period [27]. The volatilization dynamics during the experimental period revealed an initial increase in the rate, which was more pronounced in wet weather conditions, particularly during the first five days (Fig 2A). In contrast, for the dry conditions, the initial rise persisted for two days before gradually declining throughout the experimental period (Fig 2A). Saturated soil can increase the hydrolysis of urea, but reduce porosity due to water filled pores can potentially increase the $NH_3$ volatilization [28]. Previous studies have demonstrated that volatilization can persist for 1 to 9 days, depending on the soil condition and the source of the fertilizer [29,30]. However, losses can be more prolonged, particularly when the initial soil moisture content is high [23,24]. To reduce $NH_3$ volatilization from UAN, it is essential to apply the fertilizer when soil and air temperatures are cool, the soil surface is moderately dry, or when rain occurs soon after application. Rain helps in incorporating the fertilizer into the soil shortly after application and can also significantly reduce or prevent $NH_3$ volatilization. However, pursuing rain can pose its challenges. Urease inhibitors can be used with UAN to effectively reduce $NH_3$ volatilization losses [31].

The $NH_4$-N produced through hydrolysis is further transformed into $NO_3^-$ in a two-step process known as nitrification. Initially, $NH_3$ is oxidized to nitrite ($NO_2^-$) in a process called $NH_3$ oxidation. Subsequently, $NO_2^-$ is further oxidized to $NO_3$ in a process known as nitrite oxidation. Factors such as compaction, moisture, and their interaction with soil type have a significant impact on nitrification rates (Table 1). In sandy loam soil for the control treatment under dry weather, elevated residual $NH_4$-N levels were observed, particularly at depths exceeding 10 cm (Fig 3). This can be partly explained by the soil's higher pH of 8.0 and its higher CEC, influenced by its higher Ca and Mg contents. Higher CEC can enhance the retention of positive ions like $NH_4^+$. Additionally, this observation can be attributed to the frequent addition of minimum water to the surface layer (**S2 Table**), which increases nitrification at the surface but not in the lower layers due to decreased water flow and minimal percolation. Conversely, wet conditions in sandy loam soil result in greater water percolation, leading to reduced residual $NH_4$-N concentrations. Low soil water content can limit nitrifying bacterial activity by restricting substrate supply and causing dehydration [32,33]. Loam soil possesses clay particles capable of absorbing and retaining moisture over extended periods, thereby supporting microbial growth and nitrification processes that depend on moisture [34]. Consequently, minimal residual $NH_4$-N was detected in loam soil profiles under controlled conditions. Compaction can significantly reduce the rate of nitrification and increase residual $NH_4$-N in the soil. Surface compaction, occurring in the top ~10 cm, caused a reduction in nitrification rates beneath the compacted layer especially in a dry environment, implying insufficient water percolation to support microbial nitrification processes (Fig 3). The effect is

more pronounced in sandy loam soil due to its increased susceptibility to compaction [35]. Higher residual $NH_4$-N concentrations were observed in soil profiles deeper than 14 cm with SSC (Fig 3). Interestingly, higher residual $NH_4$-N levels were found in loam soils compared to sandy loam soils. Loam soils have lower infiltration rates and greater water retention capacity, allowing the top layers in SSC to maintain enough moisture to support nitrification. However, reduced water flow and aeration rates below the compacted layer significantly decreased nitrification rates. This phenomenon is especially prominent in dry environments compared to wet ones. Studies have reported drainage can be a major factor in determining the nitrification and accumulation of $NO_3$-N [36]. In dry conditions, reduced moisture percolation beneath sub-surface compacted layers in loam soil substantially lowered nitrification rates and increases residual $NH_4$-N levels. In wet soil, residual $NH_4$-N concentrations are lower than those in dry conditions, but the impact of compaction on nitrification rates remains. The infiltration rate of sandy loam soil is higher than that of loam soil, and sub-surface compaction shows minimal differences when compared to loam soil. However, the effect of compaction persists, especially in dry weather (Fig 3). A study by Whisler et al., (1965) [37] has also indicated an increase in soil compaction increases the amount of residual $NH_4$-N and decreases the recoverable $NO_3$-N from the soil profile. Similar findings were also observed in a study by Longepierre et al., (2022) [13].

The interaction between compaction and soil types significantly influenced the residual $NO_3$-N concentration within the soil profile. Both control and SC retained elevated residual $NO_3$-N levels in loam soil. This observation is consistent and corresponds with the results from residual $NH_4$-N, as minimal to negligible residual $NH_4$-N was detected in loam soil under control conditions. In sandy loam soil under control conditions, minimal residual $NO_3$-N was observed throughout the profile (Fig 4), in contrast to the higher residual $NH_4$-N levels (Fig 3). Residual $NO_3$-N in SSC did not exhibit any statistically significant differences across the soil profile for both loam and sandy loam soil (Fig 4). The highest residual $NO_3$-N concentration was recorded at 24 cm in the control treatment for loam soil. Correspondingly, increased $NO_3$-N leaching was observed in loam soil compared to sandy loam and in wet weather conditions. The rate of water flow can vary depending on compaction and soil types and can influence the nitrification rate and $NO_3$-N mobility throughout the soil profile. Wet weather conditions augment water infiltration rates, potentially resulting in increased $NO_3$-N leaching as excess water transports dissolved $NO_3$-N deeper into the soil profile. Nitrate tends to leach more rapidly from sandy soils than from finer-textured soils due to the lower water-holding capacity of sandy soils [38]. However, in the present study, higher leaching was observed in loam soil, which might be attributed to the elevated rate of nitrification in loam soil compared to sandy loam, as well as the initial higher amount of residual $NO_3$-N present in loam soil. Studies have reported higher nitrification rates in loam soils compared to sandy loam soils [39,40]. Compaction did not have any significant effect on $NO_3$-N leaching, but comparatively higher leaching was observed in surface compaction (Table 1). Nitrate leaching in wet weather occurs when heavy rainfall causes $NO_3$-N to migrate downward in the soil, below the root zone, rendering them inaccessible to plants. The likelihood of $NO_3$-N continuing to leach downward and into groundwater depends on the underlying soil and bedrock conditions as well as groundwater depth. The leaching impact not only depends on the volume of rainfall but also the amount of $NO_3$-N present in the soil. The higher rate of nitrification and increased residual $NO_3$-N significantly determined the elevated leaching potential of loam soil.

Denitrification and $N_2O$ emissions are influenced by various factors, such as soil moisture, temperature, microbial activity, aeration, and organic matter content [41,42]. Recent findings suggest increased risks of $N_2O$ emissions with intensified drying and wetting conditions due to climate-induced soil moisture variability [43]. Nitrous oxide is primarily produced during

the microbial process of denitrification, in which $NO_3$-N is converted to nitrogen ($N_2$). Nitrous oxide emissions were significantly impacted by the interaction of compaction * soil type and compaction * moisture (Tables 1 and 2). The highest $N_2O$ emissions were recorded in the control treatment combined with loam soil, while a similar pattern was observed in the control treatment under dry conditions (Table 2). The higher rate of nitrification in loam soil describes the higher potential of $N_2O$ emission from the loam soil. Denitrification was more significant in dry weather. Reduced nitrification rate in compaction treatment reduced the $N_2O$ emission. However, no significant distinctions were detected between the effects of SC and SSC and their interactions with soil types and moisture levels on $N_2O$ emission (Table 2). This implies that although compaction does influence $N_2O$ emissions, the specific characteristics of the compaction (surface vs. sub-surface) and its interaction with soil type and moisture may not play a critical role in determining emission levels. The finding contradicts the current observations of compaction increases $N_2O$ emissions [44]. The small-scale setup of the experiment might be a critical factor in the contradictory results. Other reasons such as reduced rate of nitrification due to slow water movement, aeration, and accumulation of higher $NH_4$-N in the compacted soil might have reduced the $N_2O$ emission [13,37]. Compaction also reduces the nitrifiers bacteria and increases potential diazotrophs, which might have reduced the rate of nitrification and reduced the emission [13]. This also indicates not only physical parameters, but biological indicators should be considered in truly assessing the N-dynamics in compacted soil. A summarized pathway between compaction, soil types, and moisture is provided in Fig 5.

## 5. Limitations of the study

This research was carried out in a controlled laboratory environment using soil columns, an approach that may not fully replicate the complexities of actual field conditions. Additionally, the absence of plants in our column experiments means that important interactions, such as plant uptake of $NO_3$ and its effects on the soil microbial community, were not accounted for. This omission could limit the applicability of our findings to real-world agricultural settings where plant dynamics play a crucial role. Furthermore, the study's duration was limited to 30 days, which, while providing initial insights, is not sufficient to capture the long-term effects and interactions of compaction, soil types, and moisture on nitrogen dynamics. A more extended study period could yield a deeper and more comprehensive understanding of these interactions over time. Despite these limitations, the findings of this study offer valuable insights into the initial stages of N dynamics under varying soil types, compaction levels, and moisture conditions. The controlled laboratory setting allowed for a detailed examination of specific processes and interactions that are difficult to isolate in field conditions. Therefore, while acknowledging these constraints, the results of this study still hold significant relevance and provide a foundational understanding that can inform further research and practical applications in soil and agricultural science.

## 6. Recommendations

Crop producers should exercise diligence when applying N-based fertilizers. Broadcast application (surface spray) of UAN in saturated or moist soil can result in higher N loss through ammonia volatilization. It is recommended to apply fertilizer under moderately dry soil conditions to mitigate N loss. Loam soil exhibits increased nitrification efficiency owing to its higher water retention capacity and organic matter content. For loamy soil textures with high organic matter, it is recommended to recalibrate the fertilizer rate accounting the N credit from organic matter. Additionally, implementing a split application of N fertilizer for such soil is advised to optimize nutrient uptake and minimize losses. Furthermore, higher nitrification in

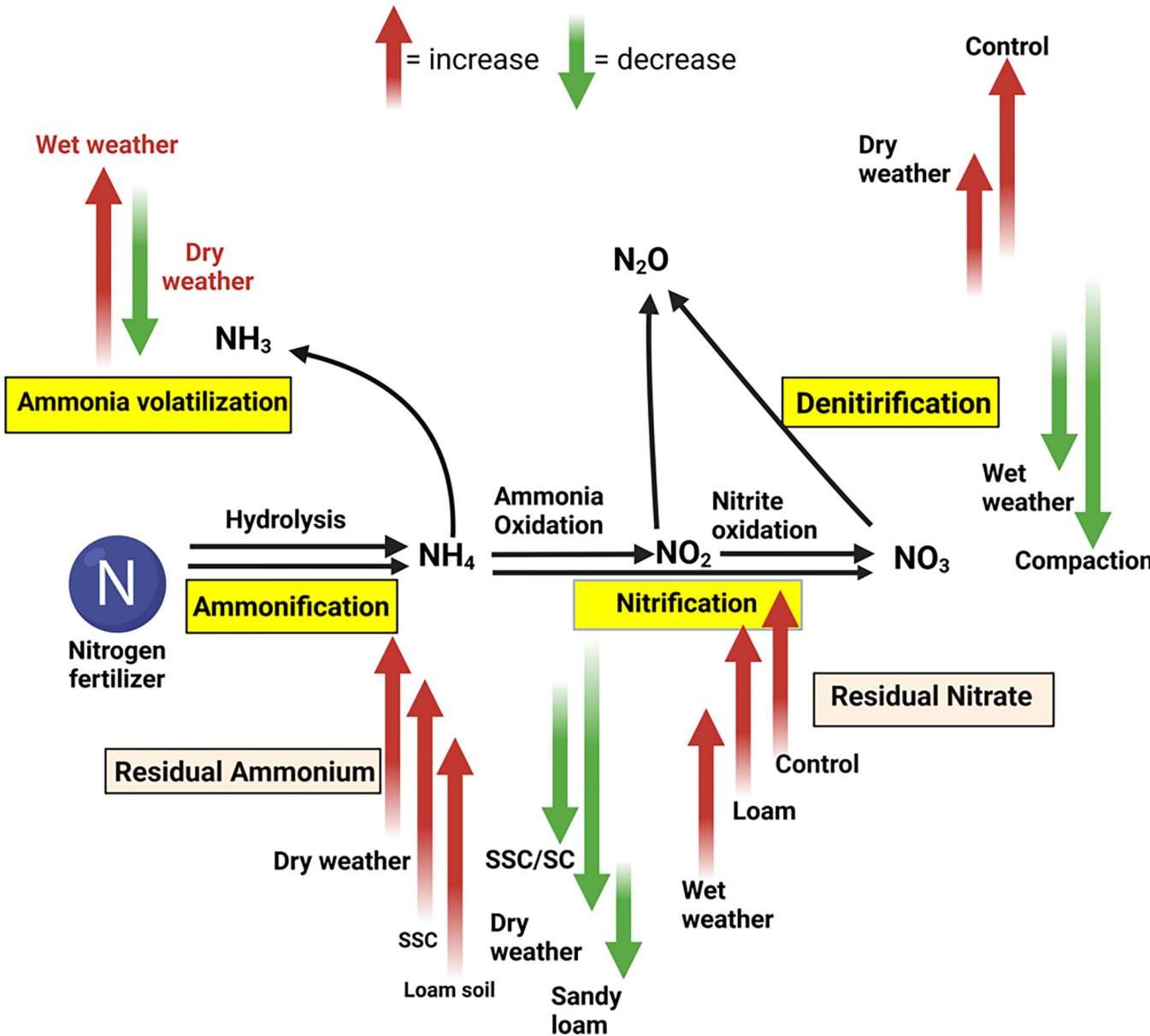

**Fig 5. Summarized results in graphical abstract.** Here red arrow represents an increase in particular nitrogen cycle steps, while the green arrow indicates a decrease; SC = surface compaction, and SSC = sub-surface compaction. Here, the control represents "no compaction" treatment.

loam soil can contribute to higher $N_2O$ emissions, a potent greenhouse gas. Employing a split application approach can substantially reduce potential $N_2O$ emissions. Soil compaction can significantly reduce nitrification rates by constraining water percolation through the soil profile. As a result, compaction leads to increased residual $NH_4$-N concentrations within the soil profile, which may negatively impact plant growth and yield, particularly under arid climatic conditions.

## Supporting information

**S1 Table. Baseline properties of the soil used in the experiment.**
(DOCX)

**S2 Table. Precipitation regime used for this study.**
(DOCX)

## Author Contributions

**Conceptualization:** Saurav Das, Bijesh Maharjan.

**Data curation:** Saurav Das, Ankita Mohapatra, Karubakee Sahu, Dinesh Panday, Deepak Ghimire, Bijesh Maharjan.

**Formal analysis:** Saurav Das, Ankita Mohapatra, Karubakee Sahu, Dinesh Panday, Deepak Ghimire, Bijesh Maharjan.

**Methodology:** Ankita Mohapatra, Karubakee Sahu, Dinesh Panday.

**Resources:** Dinesh Panday.

**Software:** Saurav Das.

**Supervision:** Saurav Das.

**Validation:** Bijesh Maharjan.

**Visualization:** Saurav Das.

**Writing – original draft:** Saurav Das, Ankita Mohapatra, Karubakee Sahu, Dinesh Panday, Deepak Ghimire, Bijesh Maharjan.

**Writing – review & editing:** Saurav Das, Ankita Mohapatra, Karubakee Sahu, Dinesh Panday, Bijesh Maharjan.

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
