## [Decision Letter · Decision Letter 0]

5 Feb 2024

PONE-D-23-37015Nitrogen dynamics as a function of soil types, compaction, and moisturePLOS ONE

Dear Dr. Das,

Thank you for submitting your manuscript to PLOS ONE. After careful consideration, we feel that it has merit but does not fully meet PLOS ONE’s publication criteria as it currently stands. Therefore, we invite you to submit a revised version of the manuscript that addresses the points raised during the review process.

**The manuscript has merit but would need significant revision.  **

We look forward to receiving your revised manuscript.

Kind regards,

Rishi Prasad, Ph.D.

Academic Editor

PLOS ONE

Journal Requirements:

Whilst you may use any professional scientific editing service of your choice, PLOS has partnered with both American Journal Experts (AJE) and Editage to provide discounted services to PLOS authors. Both organizations have experience helping authors meet PLOS guidelines and can provide language editing, translation, manuscript formatting, and figure formatting to ensure your manuscript meets our submission guidelines. To take advantage of our partnership with AJE, visit the AJE website (http://aje.com/go/plos) for a 15% discount off AJE services. To take advantage of our partnership with Editage, visit the Editage website (www.editage.com) and enter referral code PLOSEDIT for a 15% discount off Editage services. If the PLOS editorial team finds any language issues in text that either AJE or Editage has edited, the service provider will re-edit the text for free.

4. In the online submission form, you indicated that Data can be provided on request, as it is currently in use for other studies.

Reviewers' comments:

Reviewer's Responses to Questions

**Comments to the Author**

1. Is the manuscript technically sound, and do the data support the conclusions?

Reviewer #1: Yes

Reviewer #2: Yes

Reviewer #3: Yes

2. Has the statistical analysis been performed appropriately and rigorously? 

Reviewer #1: Yes

Reviewer #2: Yes

Reviewer #3: Yes

3. Have the authors made all data underlying the findings in their manuscript fully available?

Reviewer #1: Yes

Reviewer #2: Yes

Reviewer #3: Yes

4. Is the manuscript presented in an intelligible fashion and written in standard English?

Reviewer #1: Yes

Reviewer #2: Yes

Reviewer #3: Yes

5. Review Comments to the Author

Reviewer #1: This is a well written paper with potentially useful dataset. The experimental design and data collection procedures appear to be good. This work can be accepted for publication after accurate minor revisions.

Reviewer #2: The authors presented research work on nitrogen dynamics in soil, which is an important issue for cropping systems, and examined experimental laboratory methods to support the subject topics. I thoroughly enjoyed reviewing the work. The article is well-written and contains valuable outcomes that would help future researchers in this domain. Before accepting the manuscript, I would like to suggest addressing my comments given below:

I have a major comment:

Line 41-106: The introduction section needs to be improved as soil type and soil moisture functions were missing as major contributors to N-dynamics. Most of the text concerns soil compaction, which doesn’t clearly expose the whole scenario of the research topic.

Besides them, I have some minor comments:

Line 13–38: It would be better to include the important statistical values that can enhance the robustness of the findings and provide a clearer understanding of the study's reliability.

Line 41–106: The transition from the general context to the specific study objectives is somewhat abrupt. Consider providing a more seamless transition that explicitly connects the broader issues discussed to the rationale for the present study.

Line 49–71: Some sentences are quite lengthy and complex. Consider breaking down complex sentences into shorter, more digestible ones for improved clarity and readability.

Line 110: I was wondering why the author chose a 0–20 cm soil depth for the experiment.

Lines 117–119: Please explain the rationale behind maintaining the gravimetric water content at 10%, specifying why 70% and 50% of field capacities were chosen for sandy loam and loam, respectively, in this study.

Line 172: Need to update the reference “Mosier and Mack, 1980” with a more recent reference

Line 284–291: Please add references to support the statements.

Line 292-298: Better to update the reference “Bouwmeester et al., 1985” with a more recent reference

Line 325–329: Please add references to support the statements.

Line 344–347: Please add references to support the statements.

Line 292-298: Need to update the reference “Whisler et al., 1965” with a more recent reference

Lines 415 and 418: Better to use “N2O” than “nitrous oxide”

In general, it would be better to add a small section of “limitations of the study” and clear “recommendations for future researchers” on how the outcomes of this work would help foster sustainable agriculture.

Reviewer #3: Please see the attached file.

6. PLOS authors have the option to publish the peer review history of their article (what does this mean?). If published, this will include your full peer review and any attached files.

Reviewer #1: No

Reviewer #2: No

Reviewer #3: No

---

## [Author Response · Author response to Decision Letter 0]

1 Mar 2024

Response to reviewers’ comments

(all the responses are in red italic font)

Reviewer 1

Comments to the Author:

Line 17: I would put soil types (loam and sandy loam) in parentheses.

Response: Revised accordingly.

Line 34: Here, ‘moist soil’ might be too broad, as you have only potentially shown this outcome in sandy

loam and loam soil, not all types of soil.

Response: Revised accordingly

Line 38: Lowercase O in the word ‘Oxide’.

Response: Revised accordingly.

Line 41: Remove space before paragraphs. Keep the format consistent throughout the manuscript.

Response: Revised accordingly.

Line 53-54: This sentence is partially true. Denitrification and volatilization are just one of the factors,

contributing to global warming, as carbon dioxide accounts for a larger proportion.

Response: Revised accordingly, added “among” the primary contributors of N losses and global warming. 

Line 54-56: Add references.

Response: Reference added. 

Line 82-85: Add references and explain how this subsequently affects N transformation and movement.

Response: Reference added. 

Line 71-100: The title of this manuscript is ‘nitrogen dynamics as a function of soil types, compaction,

and moisture’. Here only background information of compaction was introduced, it would be good if

some background information of used soil types and moisture can be added.

Response: An introductory section on soil types and moisture and its interplay on N transformation has been added. 

Line 131: Compaction affects N dynamics. Can you explain how you pack sandy loam soil into columns with a targeted bulk density of 1.6 kg/m3 on the top 10 cm depth? Will the bulk density of the bottom 10 cm depth soil be affected during this packing process?

Response: We measured the soil needed for the bottom 10 cm and the top 10 cm based on the targeted compaction and accordingly added the required amount; we used considerable care while packing the soil.

Line 149-153: Water was added daily; please give an example of how much water content (in mL) was

added per day?

Response: A supplementary sheet (Supplementary file S2) was added showing the amount of water added daily. 

Line 155: Lowercase C in the word ‘collection.’ Remove space before paragraphs.

Response: Revised accordingly. Space was removed from the paragraph.

Line 161: Can you explain how you determine the time interval and what is the threshold for each sponge to absorb ammonia?

Response: In many cases, N losses through volatilization primarily happen within the initial seven to fourteen days following application. As observations progress, these losses generally decrease significantly. For our study, we conducted daily observations for the first three days, then alternated every other day up to the 21st day, and subsequently extended the intervals to four days. This information is included in the revised manuscript. 

We did not establish a threshold level for each sponge to absorb ammonia but instead harvested everything when it was time to change the sponge.

Line 164-165: Lowercase K in the word ‘kg.’

Response: revised accordingly.

Linr166-168: The volume of N2O gas for analysis and calculation is confusing. Please explain it and show the calculation.

Response: The calculation of N2O gas has been added to the materials and methods. 

Line 183: Check the hyphen in NH4-N and NO3-N.

Response: Revised accordingly

Line 187: This section should be 2.4, not 2.5. Lowercase A in the word ‘Analysis’. Remove space before

paragraph.

Response: Revised accordingly.

Line 217: lowercase A in the word ‘ammonia’.

Response: Revised accordingly.

Line 228: There’s no difference in N2O emission among all treatments except the C x D treatment.

Response: Explained

Line 238-239: ‘both exhibited similar patterns of gas emissions’ is confusing. This is not true for NH3 and

N2O. Please explain it.

Response: Revised, and we have removed that sentence. 

Line 240: The trend you described here is correct if it’s NH3 volatilization value, but not for NH3

volatilization rate.

Response: Revised accordingly.

Line 246-247: Add space between paragraphs.

Response: Revised accordingly.

Line 257: The lowest was observed in wet weather under control treatment (here should be CxW, not

CxD). The lowest was also observed in SCxW treatment, there’s no difference in residual NH4-N

between SCxW and CxW.

Response: Corrected. 

Line 259: 4 in ‘NH4-N’ should be the subscript.

Response: Revised accordingly.

Line 262: Lowercase K in the word ‘kg’.

Response: Revised accordingly.

Line 323: Change the reference format of BEAUCHAMP et al., 1982 and check other citations’ format

throughout the manuscript.

Response: Revised accordingly.

Line 324-325: Add references; how much initial soil moisture content will be considered high?

Response: Revised accordingly. 

Line 327-329: Will rain cause more fertilizer losses through surface runoff compared to reducing NH3

volatilization?

Response: While increased rainfall could potentially lead to more fertilizer losses through surface runoff, this scenario was not observed in our simulated study given the specific precipitation levels replicated. However, it is entirely plausible under different conditions where heavier or more frequent rainfall might occur.

Line 344-346: Loam soil possesses clay particles……; sandy loam could have the same amount of clay

content; please specify the percentages of clay, silt, and sand for the two types of soil in this study.

Response: data provided in the material and method section. [section 2.1: description of soil]

Line 369: Remove comma.

Response: Revised accordingly.

Line 386-387: In the word ‘NO3-N’, 3 should be the subscript.

Response: Revised accordingly.

Line 387-389: Add references.

Response: Reference added. 

Line 441: For those soil with high organic matter and residual nitrate, why not suggest using less amount.

of fertilizer?

Response: Thank you for this comment; we have revised and added our recommendation.

Figure 2: In the results part, you mentioned Figures 2A and 2B, alphabet ‘A’ and ‘B’ should also show in

each figure.

Response: Revised accordingly.

Figures 3 and 4: Lowercase K in the word ‘kg.’

Response: Revised accordingly.

Table 1: The standard deviations for leachate and soil residual mineral N are relatively high, so I’m

wondering if this experiment is repeatable or not?

Response: Thank you for this comment. We acknowledge this aspect and limitation of the study, which primarily focuses on illustrating the general process of nitrogen dynamics in relation to different soil types, compaction levels, and moisture conditions. The aim of this research was to provide a broad understanding of the dynamics.

Table 2: P value in this table is confusing, I would put p value at the bottom of each section instead of top.

Response: Revised accordingly.

Symbol ‘±’ is missing in terms of residual NH4-N under SCxW treatment.

Response: Revised accordingly.

Reviewer 2

The authors presented research work on nitrogen dynamics in soil, which is an important issue for cropping systems, and examined experimental laboratory methods to support the subject topics. I thoroughly enjoyed reviewing the work. The article is well-written and contains valuable outcomes that would help future researchers in this domain. Before accepting the manuscript, I would like to suggest addressing my comments given below:

I have a major comment:

Line 41-106: The introduction section needs to be improved as soil type and soil moisture functions were missing as major contributors to N-dynamics. Most of the text concerns soil compaction, which doesn’t clearly expose the whole scenario of the research topic.

Response: A section discussing soil type and moisture has been added to the introduction. 

Besides them, I have some minor comments:

Line 13–38: It would be better to include the important statistical values that can enhance the robustness of the findings and provide a clearer understanding of the study's reliability.

Response: Revised accordingly.

Line 41–106: The transition from the general context to the specific study objectives is somewhat abrupt. Consider providing a more seamless transition that explicitly connects the broader issues discussed to the rationale for the present study.

Response: Revised to improve the flow of the text. 

Line 49–71: Some sentences are quite lengthy and complex. Consider breaking down complex sentences into shorter, more digestible ones for improved clarity and readability.

Response: Revised accordingly.

Line 110: I was wondering why the author chose a 0–20 cm soil depth for the experiment.

Response: In our simulated laboratory study, we chose the 0 to 20 cm soil layer to replicate the plow layer and the soil moisture conditions in the field.

Lines 117–119: Please explain the rationale behind maintaining the gravimetric water content at 10%, specifying why 70% and 50% of field capacities were chosen for sandy loam and loam, respectively, in this study.

Response: Thank you so much for your comment. This has been revised accordingly in the manuscript.

Line 172: Need to update the reference “Mosier and Mack, 1980” with a more recent reference

Response: A new reference has been added.

Line 284–291: Please add references to support the statements.

Response: This is the study's primary objective, and the sentence is more based on the study's results. 

Line 292-298: Better to update the reference “Bouwmeester et al., 1985” with a more recent reference

Response: Revised accordingly.

Line 325–329: Please add references to support the statements.

Response: Revised accordingly.

Line 344–347: Please add references to support the statements.

Response: Revised accordingly.

Line 369: Need to update the reference “Whisler et al., 1965” with a more recent reference

Response: This was one of the first studies on soil compaction and N transformation; we have added one more reference along with this.

Lines 415 and 418: Better to use “N2O” than “nitrous oxide”

Response: Revised accordingly throughout the manuscript. 

In general, it would be better to add a small section on “limitations of the study” and clear “recommendations for future researchers” on how the outcomes of this work would help foster sustainable agriculture.

Response: A section 5.0 has been added, discussing the limitations of this study.

Reviewer 3 

The authors have conducted the present study on “Nitrogen dynamics as a function of soil types, compaction, and moisture,” which is of practical significance in view of the low use efficiency of N. The experiment has been designed well. A few comments are given below for further improvement. 

Abstract: The authors have not studied the method of application (broadcasting/ split) of fertilizer in the soil and its effect on N-dynamics in the present study. Hence, the statement in connection with the split application may be deleted.

Response: Revised accordingly. 

Introduction: 

The authors emphasized soil compaction and moisture content while formulating the hypothesis. However, it does not emphasize soil types or textural composition. The research gap is poorly defined. The novelty in the experimental setup (simulated condition) in carrying out studies on N dynamics may be highlighted.

Response: Thank you for this comment, we have added a section on soil types and moisture in the introduction section. 

Materials and methods 

USDA soil Taxonomy may be used against the soils used in the study.

Response: Revised accordingly, the soil series name has been added in parentheses.

Description of the sampling sites' climatic condition (data) may be included in this section, since the sampling sites vary in climatic condition.

Response: Revised accordingly.

Details of application method (e.g. spray) of UAN in soil to be highlighted.

Response: Revised accordingly. 

Why was the N dose equivalent to 200 kg N/ha applied? What is the basis? 

Response: Explained accordingly.

The methodologies followed to characterize the soil for its properties may be included in this section. For the micronutrients, clearly indicate the extractant(s) used for the estimation of the nutrients in the soil.

Response: Revised accordingly.

The potassium contents of soils are very high; the reason may be given. The Olsen P in loamy soil seems to be high. The data may be rechecked.

Response: We have re-examined our data and confirmed its accuracy. The high potassium contents in the soils might be attributed to factors such as historical fertilizer applications, which could have included potassium or phosphorus supplements. Unfortunately, detailed records of these past applications were not available for our analysis. Additionally, the loamy soil's higher organic matter content is likely contributing to elevated Olsen P levels. Organic matter can enhance phosphorus availability and retention in soil. Thus, while we acknowledge these factors as potential contributors to the observed nutrient levels, the exact historical agricultural practices remain uncertain. Our findings, therefore, reflect the current state of the soil under study, considering these possible influences. 

Results 

The reason behind the higher leaching of nitrate (NO3-) under loamy soil than sandy loam soil (with higher sand percentage) may be indicated.

Response: The reason has been explained and discussed in the “Discussion” section with line number 443– 446. 

“However, in the present study, higher leaching was observed in loam soil, which might be attributed to the elevated rate of nitrification in loam soil compared to sandy loam, as well as the initial higher amount of residual NO3-N present in loam soil. Studies have reported higher nitrification rates in loam soils compared to sandy loam soils ….”

Line no. 226-229: highest N2O emission was recorded in control treatment in dry weather (C x D), and the effects of other treatments were at par with that in SSC x D. please rectify. 

Response: Revised accordingly.

Residual nitrate was significantly lower in sandy loam soil… which indicates higher leaching in sandy loam (since the mean effect of soil type on other losses was non-significant). But the data indicates the reverse.

Response: In response to your observation about the lower residual nitrate in sandy loam soil, we have addressed this point in the "Discussion" section of our paper. Specifically, we found that the sandy loam soil exhibited a higher concentration of residual ammonium (NH4-N). This implies that the nitrification process was less active in sandy loam compared to loam. As a result, despite the initially higher potential for leaching in sandy loam, the overall nitrogen balance is maintained due to this reduced rate of nitrification. This factor contributes to the observed discrepancy in the data and helps explain the overall nitrogen dynamics in the sandy loam soil.

Why, under compacted conditions, residual nitrate was reduced in spite of the fact that nitrous oxide emission was significantly lower under compacted conditions. 

Response: The observed reduction in residual nitrate under compacted conditions, alongside lower nitrous oxide emissions, can primarily be attributed to the impact of compaction on soil aeration and microbial activity. Compaction reduces soil porosity, limiting oxygen availability and thus impeding the nitrification process, which is essential for converting ammonium to nitrate. This results in lower nitrate levels. Simultaneously, the limited availability of nitrate and altered soil conditions under compaction also affect the denitrification process, potentially leading to lower nitrous oxide emissions. 

Rectify lines 331-333. 

Response: Revised accordingly.

In case of control and surface compact condition under dry and wet situation, residual ammonium was reported lower in loamy soil in comparison to sandy loam soil. The author did not consider the cation exchange capacity (CEC) of soils. It is reported in the study that the CEC of sandy loam soil is higher in-spite of the fact that organic carbon content of the soil is lower than that of the loamy soil. Cation exchange capacity of soil has significant positive impact on the retention of positively charged ammonium ion (NH4+) in soil.

Response: Thank you for your insightful comment. In our revised manuscript, we have now included a more comprehensive discussion considering the soil properties you've highlighted. The sandy loam soil, with its higher pH of 8.0, does indeed influence the Cation Exchange Capacity (CEC). This alkaline pH can enhance the availability of cationic exchange sites, as more hydrogen ions are displaced at higher pH levels, potentially increasing the CEC. This factor could contribute to the improved retention of NH4+ in the sandy loam soil. Additionally, the higher content of calcium (Ca) and magnesium (Mg) in the sandy loam, despite its lower organic matter and organic carbon content compared to the loamy soil, might also play a role in its CEC. 

Recommendation paragraph 

The authors have not recorded the soil or air temperature. Hence, the sentence in connection with soil and air temperature may be deleted from the recommendation paragraph. 

Response: Revised accordingly.

The manuscript may be accepted for publication following major revision.

Thank you.

---

## [Editor Report · Decision Letter 1]

14 Mar 2024

Nitrogen dynamics as a function of soil types, compaction, and moisture

PONE-D-23-37015R1

Dear Dr. Das,

We’re pleased to inform you that your manuscript has been judged scientifically suitable for publication and will be formally accepted for publication once it meets all outstanding technical requirements.

Kind regards,

Rishi Prasad, Ph.D.

Academic Editor

PLOS ONE
---

## [Editor Report · Acceptance letter]

26 Mar 2024

PONE-D-23-37015R1 

PLOS ONE

Dear Dr. Das, 

I'm pleased to inform you that your manuscript has been deemed suitable for publication in PLOS ONE. Congratulations! Your manuscript is now being handed over to our production team.

Kind regards, 

on behalf of

Dr. Rishi Prasad 

Academic Editor

PLOS ONE